# Technical Assessment of Ultrasonic Cerebral Tomosphygmography and New Scientific Evaluation of Its Clinical Interest for the Diagnosis of Electrohypersensitivity and Multiple Chemical Sensitivity

**DOI:** 10.3390/diagnostics10060427

**Published:** 2020-06-24

**Authors:** Frédéric Greco

**Affiliations:** Anesthesia and Intensive Care Unit, University Hospital of Montpellier, F-34295 Montpellier, France; F-greco@chu-montpellier.fr

**Keywords:** electrohypersensitivity, multiple chemical sensitivity, ultrasonic cerebral tomosphygmography, encephaloscan, pulsed ultrasounds, pulsatile echoencephalography, diagnosis

## Abstract

Ultrasonic cerebral tomosphygmography (UCTS), also known as “encephaloscan”, is an ultrasound-based pulsatile echoencephalography for both functional and anatomical brain imaging investigations. Compared to classical imaging, UCTS makes it possible to locate precisely the spontaneous brain tissue pulsations that occur naturally in temporal lobes. Scientific publications have recently validated the scientific interest of UCTS technique but clinical use and industrial development of this ancient brain imaging technique has been stopped notably in France, not for scientific or technical reasons but due to a lack of financing support. UCTS should be fundamentally distinguished from transcranial Doppler ultrasonography (TDU), which, although it also uses pulsed ultrasounds, aims at studying the velocity of blood flow (hemodynamics) in the cerebral arteries by using Doppler effect, especially in the middle cerebral artery of both hemispheres. Instead, UCTS has the technical advantage of measuring and locating spontaneous brain tissue pulsations in temporal lobes. Recent scientific work has shown the possibility to make an objective diagnosis of electrohypersensitivity (EHS) and multiple chemical sensitivity (MCS) by using UCTS, in conjunction with TDU investigation and the detection of several biomarkers in the peripheral blood and urine of the patients. In this paper, we independently confirm the clinical interest of using UCTS for the diagnosis of EHS and MCS. Moreover, it has been shown that repetitive use of UCTS in EHS and/or MCS patients can contribute to the objective assessment of their therapeutic follow-up. Since classical CT scan and MRI are usually not contributive for the diagnosis and are poorly tolerated by these patients, UCTS should therefore be considered as one of the best imaging technique to be used for the diagnosis of these new disorders and the follow-up of patients.

## 1. Introduction

Electrohypersensitivity (EHS) is a new World Health Organization (WHO)-acknowledged disabling condition occurring in EHS self-reporting patients [1]. However, because electromagnetic fields (EMF) exposure was not proven to cause EHS, instead of using the term electrohypersensitivity, it was proposed to use the term “idiopathic environmental intolerance (IEI) attributed to electromagnetic fields (IEI-EMF)” to qualify this new EHS-Associated pathology [2]. It has been recently reported that, in addition to peripheral blood and urine biomarker detection, ultrasonic cerebral tomosphygmography (UCTS) can be used to diagnose EHS and multiple chemical sensitivity (MCS) [3,4,5,6]. The clinical use of UCTS in EHS and/or MCS patients has been recently criticized by the French Agency for Food, Environmental and Occupational Health & Safety (ANSES, Maisons-Alfort, France) [7], prompting us to make an independent expert assessment by: (1) recalling what precisely UCTS is and how it works technically; (2) briefly reviewing the medical literature to evaluate the physiopathological meaning and potential clinical interest of this brain imaging technique; and, more specifically, (3) questioning whether it can actually contribute to the diagnosis of EHS and MCS.

## 2. Scientific Background

First, we have to recall that UCTS is a pulsatile ultrasound-based echography analyzing brain tissue that was introduced for the first time in France in 1966 [8], “sphygmo” meaning in Greek pulse and “sphygmography” the technique to record it.

In fact, after the identification of ultrasounds and use of echoencephalography in 1956 by L. Leksell [9], numerous international scientific publications have confirmed the existence of a spontaneous natural cerebral pulsatility that is synchronous with cardiac systoles [10,11,12,13,14,15,16,17,18,19,20,21,22,23].

Detection of cerebral pulsations by using pulsatile echoencephalography was particularly well grasped by Canadian researcher D.N. White in two peer-reviewed scientific papers published in 1980 and 1992, respectively [24,25], citing French works on UCTS. To this, we should add a dozen French medical doctoral thesis from 1983 to 1994 and, more recently, several PhDs in science that back-up the scientific validity of the method, and confirm that cerebral pulsation measurement can be recorded in standardized and reproducible conditions. Note that these pioneering works were revisited notably by the contribution of M.E. Wagshul et al. in 2011 [23].

Consequently, the large amount of past and more recent scientific work opposes those who claim that echoencephalography (including UCTS) is an obsolete method of cerebral investigation with no clinical interest.

## 3. What Is UCTS?

As indicated above, UCTS is a pulsatile ultrasound echoencephalographic method using, as its name suggests, the emission–reception of intermittent pulsed ultrasound, which allows real-time measurement of spontaneous pulsations of the neurovascular cerebral tissue, region by region in temporal lobes. It is therefore a non-invasive “echotomoencephalographic” method leading to both functional and anatomical cerebral tissue analysis, based on an “ultrasonic centimetric recording” of the natural pulsations of this tissue. This technique was originally developed in France mostly by J.M. Lepetit [18,19,20,21] who worked in the 1970s and 1980s in the Neurological Functional Exploration Laboratory at the Dupuytren University Hospital in Limoges, France, where the first echotomoencephalograph or tomosphygmograph was built. As underlined above, the originality of UCTS compared to classical pulsatile echoencephalography is that measurement and recording of the cerebral pulsations take place in different tissue regions of the temporal lobes, making it possible to locate precisely these pulsations in the different neuronal structures within these lobes, resulting in a fundamentally more sensitive and precise method for monitoring neuronal functions and capillary hemodynamics in brain tissue.

## 4. UCTS Engineering and Technical Equipment to Measure and Record Spontaneous Cerebral Pulsations

The first prototype was built in Limoges (France). It is presently no longer in use. However, a few other machines have been built. Such an apparatus still in use in 2018 is shown in Figure 1, although pulsatile echoencephalographs presumably exist in other research centers worldwide.

In the 1980s and 1990s, several other machines were built. However, their manufacturing was dropped for financial problems. This resulted in the discontinuation of the use of UCTS in medical practice, not for scientific or technical reasons, but due to a lack of available machines and to the development of B mode imaging ultrasound and transcranial Doppler.

Today, such devices can only be used in the context of research.

As indicated in Figure 2, these devices are based on the technical arrangement of several elements, which comprise the following.

Two ultrasound emitting–receiving probes are placed on each side of the head in an over-ear position (1A and 1B) and comprise the transmitter device (2), which generates voltage pulses operating in pulsed intermittent mode at a frequency of 2 MHz. The echoes reflected by the various intracerebral structures (vessel walls, cell membranes, etc.) are collected by the same probes.

These echoes are amplified by an amplifier (3) whose gain increases with depth, so as to compensate the tissue attenuation related to the absorption phenomena in the tissue they go through. This compensation is generated by a TGC (“time gain compensation”) device (4).

The amplitude of signals is then converted into binary values using a conversion system operating at a frequency of 18.8 MHz (5).

An ultrafast computing unit (6) determines the average value of the echography signal during the times corresponding to each of the 1-cm thick slice sections. Calculations take place at each wave transmission and reception cycle every 5 ms, i.e., 200 times per second. They give the average value of the maximum ranges corresponding to each of the eight 1-cm regions arranged sequentially from the cortex to the wall of the third ventricle of each hemisphere.

The depth at which the different section slices are located is adjustable centimeter by centimeter from the skin using a delay device (7), i.e., eight centimetric slices.

A filtering system by digital rejection of small echoes (8) makes it possible to exclude noise, while keeping an unchanged amplitude of the signal echoes. The QRS complexes of the ECG are recorded simultaneously with the ultrasound signal at the end of recording.

The pulsometric index (8) mean values are transferred at each transmission–reception cycle to a calculator (9), which saves them while displaying them on an oscilloscope (10). The calculator saves in its memory an amount of values corresponding to 80 s of examination.

As shown in Figure 3, the graphs of the collected pulsations are composed of a succession of points, each point representing the mean value of the ultrasound signal for a 1-cm-thick section. This mean value is calculated as indicated above every 5 ms.

The calculator determines over a period of 80 s for each 1-cm section the following parameters: the mean value of maximum (V_max_) and minimum (V_min_) amplitudes; the mean value of the amplitude variations, i.e., ΔV = V_max_−V_min_; the mean value of the slope of the rising edge of the pulsatile wave as determined on the graph at the point of ordinate ½(V_max_ + V_min_); and the mean values of the time separating the QRS complex from the maximum amplitude (Δt = Z_max_) and from the minimum amplitude (Δt = Z_min_) of the pulsatile waves, these values being calculated from the graphs obtained. At the end, the calculator displays the data in the form of digital values (pulsometric index) and bar graph (pictogram) (see Section 7).

## 5. Modalities of Use

### 5.1. Type of Probe and Positioning

The UCTS uses a 2-MHz pulsed emitting–receiving probe placed on the temporal window.

Before any measurement, as indicated in Figure 4, the probe needs to be placed so as to obtain the best possible definition of the ultrasonic field, that is to obtain a maximum amplitude of the echo waves, after reflection of the incident waves on the wall at the third ventricle, which is the echogenic reference region of echography A. Thus, guided by sound and sight, the pulsatile signal is viewed on an oscilloscope, as for an ultrasound A exam, helping ensure the correct position of the probe and the permanence of its positioning.

### 5.2. Calibration—Definition of the Pulsometric Index

Calibration was carried out initially with reference, on the one hand, to the complete circulatory stop, such as observed in the event of cerebral death (lack of cerebral pulse) [18] and conversely in the event of carbogen (O_2_ plus CO_2_) inhalation (peak pulsations observed) [27], and, on the other hand, to normalcy based on 143 normal individuals aged 18–85 [21]. It should be noted, however, that subsequently to this evaluation in normal individuals another calibration was performed from the study of 100 normal control individuals [5].

The echoes reflected by the brain structures are modulated in terms of frequency by the heart rate and in terms of amplitude by the ability of waves both to be absorbed by tissue and to be reflected on intracerebral structures. The recorded curves are the envelopes of the variations of the echo amplitude according to time. The signal amplitudes averaged over 1 min of recording make it possible to define a mean pulsometric index (MPI) for each of the areas explored, which so correspond to the mean value of the amplitude of recorded pulsations over 1 min in each temporal lobe region.

### 5.3. The Different Temporal Regions Explored

Two-millimeter-thick sections are recorded and summed by 5, which provides 1-cm-thick areas, each being characterized by a different MPI.

Seven contiguous centimetric areas were thus determined sequentially from the surface of the skull, i.e., Centimeters 3–9, Centimeters 1 and 2 standing for the skin and bone being devoid of any pulsatility.

In addition, as shown in Table 1, the individualized areas were grouped according to their expected vascular and neuronal particularity, based on different experimental and clinical data.

## 6. Search for Correspondence between the Anatomical Regions So Far Individualized in Temporal Lobes and the Different Neurovascular Structures They Are Associated with

One of the fundamental questions raised by the UCTS clinical use is the correspondence between the different 1-cm-thick regions individualized in the temporal lobes, and the vascular systems on which they rely on and the underlying neuronal structures they contain.

The scientific justification for grouping the different tissue temporal lobe-associated territories so far individualized and their correspondence with the underlying temporal lobe neuronal structures is mainly based on clinical observations.

Thus, during unilateral carotid compression, the ipsilateral side has a temporary cessation of pulsatility in the territories corresponding to Centimeters 3–7, which suggests that these territories depend on the carotid system, whereas pulsatile activity persists in the territories corresponding to Centimeters 8 and 9, which suggests that these territories do not depend on this vascular system [20,21,26]. By contrast, during a disease-associated or traumatic attack of the vertebral arteries, whatever the cause, inducing its insufficient blood flow, there is a pulsatility collapse in these territories, which suggests that the territories corresponding to Centimeters 8 and 9 depend on the vertebro-basilar artery system [28].

UCTS thus makes it possible to explore the superficial layers of the brain that depend on the carotid system (Centimeters 3–7) and the deep layers that depend on the vertebro-basilar system (Centimeters 8 and 9).

It should be noted that the pulsatility of these territories corresponding to the vertebro-basilar system (territories located at Centimeters 8 and 9) represents roughly 38% of the total hemispheric pulsatility, and that this value is of the same order of magnitude as that attributed to the blood flow of the vertebro-basilar system compared to the total cerebral blood flow, the latter including the carotid blood flow [5].

Figure 3 summarizes the currently accepted model. Note that, if attributing the cortex and sub-cortex to the territories of Centimeters 3 and 4 does not a priori constitute a problem, the marking out between the superficial Sylvian region attributed to the territory of Centimeters 3–5, and the deep Sylvian region attributed to the territory of Centimeters 5–7, as well as the individualization of the region called “intermediate” at Centimeter 7, remains unclear. Furthermore, attributing the so-called intermediate region to capsulothalamic structures remains to be confirmed. However, this region seems to be particularly important from a clinical and physio-pathologic viewpoint, as it would appear to be associated with the limbic system and the thalamus.

## 7. Presentation of Results as Pictograms

Based on the computerized recorded data obtained in normal subjects, a colored pictogram can be produced, on which appears the normal mean values and standard deviations of pulsometric index for each region of temporal lobes explored.

Table 2 shows the different values of MPIs for each region considered as determined from a series of normal control individuals.

Thus, the pictogram produced for each hemisphere shows the MPI values determined for each of the regions studied (see Figure 3) at the two temporal lobes, from the surface to the depth of the brain (Figure 4).

## 8. Physio-Pathological Relevance

As the device measures the spontaneous pulsatility of the brain tissue in the different temporal regions, the amplitude of the recorded waves—more precisely the value of the MPI in each of these regions—refers to several different parameters. Some of them are related to vascular arterial pressure and blood flow processes, such as systolic perfusion pressure (itself dependent on systemic blood pressure), cerebral blood flow, and resistance to blood flow in the arteriolo-capillary tissue networks, while others are related to the elasticity of arteriolar walls and the elasto-viscosity of the blood and brain tissue.

## 9. The UCTS Compared to Transcranial Doppler

Transcranial Doppler is fundamentally different from UCTS. Using the Doppler effect to examine intracerebral arteries hemodynamics was introduced by R. Aaslid in 1982 [29]. By directing the ultrasound beam through the temporal window, the velocity of the blood flow of the intracranial arteries, and more particularly that of the middle cerebral artery, which vascularizes approximately 60% of the brain, can be measured (Figure 5). Using this technique, the amplitude of the waves is not only measured by ultrasound after their reflection on the different intracerebral structures as is the case in pulsatile echoencephalography, but unlike UCTS, it is also the offset of wave frequencies after reflection on red blood cells moving within these arteries that is measured. In other words, the frequency of the echo waves after reflection on moving red blood cells varies, causing the immobile probe to measure the changing distance between it and the circulating red blood cells.

Thus, Doppler ultrasound makes it possible to calculate not only the pulsatility of intracerebral arteries—in particular that of the middle cerebral artery—but also the maximum and mean velocities of the blood flow, due to the existing relation between the frequency offset by Doppler effect and the blood flow velocity.

UCTS however measures pulsatility of brain tissue, and thus provides information on the local vascular state and compliance of this tissue, while transcranial Doppler provides information on vascular intracerebral hemispheric hemodynamics.

UCTS and transcranial Doppler are therefore two non-invasive functional and anatomical techniques that both use ultrasound: one analyzes the hemispheric cerebral hemodynamics, mainly that of the middle cerebral artery (transcranial Doppler), whereas the other (UCTS) locates the spontaneous pulsatility of cerebral tissue in temporal lobes.

## 10. Clinical Assessment of UCTS Use for the Diagnosis of EHS and MCS

Historically UCTS has been shown to contribute to the diagnosis of strokes, transient ischemic attacks (TIA) [30], intracerebral hematoma [26], dementia [31], and even some encephalitis. Its utility also seems to have been demonstrated in terms of carotid surgery [32,33] and in the assessment of temporo-sylvian anastomoses [34].

Globally, the relevance of using UCTS is twofold: (1) it is a non-invasive method; and (2) it can locate dysfunctional tissue early, before irreversible anatomical lesions may occur.

Nowadays, electrohypersensitivity (EHS), and so called “idiopathic environmental intolerance attributed to electromagnetic fields” (IEI-EMF) are pathological conditions recognized by WHO [1]. In fact, it has been shown that patients who claim to be EHS have a pathological condition, objectively identified by the detection of different abnormal blood biomarkers, showing cerebral low-grade inflammation [3,35] and oxidative and nitrosative stress [36].

It has been reported that, in these patients, conventional brain CT or MRI are usually normal, but that abnormalities might be found on Functional MRI sequences [37]. However, in some patients, cerebral MRI (which generates electromagnetic fields) may be poorly tolerated. As part of research work, it is therefore essential to use other brain imaging techniques to search for anomalies in the brain. Note that the type of neurological clinical symptoms presented by these patients [3,6] led to search for neuronal anomalies in the temporal lobes, e.g., by UCTS.

It has been reported that the use of UCTS in EHS- and/or MCS-bearing patients shows a decrease in cerebral pulsatility in certain territories of the temporal lobes, particularly in the capsulothalamic area which contains the limbic system and the thalamus (Figure 6) [5].

As recommended by the ANSES (the French Agency for Food, Environmental and Occupational Health & Safety) [7], we therefore decided to set up and carry out another independent analysis with the objective to verify whether these published unexpected data could be confirmed or not.

We thus considered a short series of 100 so far available identifiable EHS and/or MCS patients. Patients were randomly selected from the database used by the authors of the above publications to further analyze the results obtained by the use of UCTS in the patients.

As depicted in Table 3, we confirmed the previously reported data [5,6] by showing that in comparison with normal individuals there is a decrease in MPI, whatever the categories of patients tested (EHS, MCS, or EHS/MCS) [38]. We therefore consider that UCTS is an appropriate method to contribute to the diagnosis of EHS- and/or MCS-bearing patients.

Accordingly, future research should try to pinpoint the exact neuronal seat of the observed capsulothalamic cerebral pulsatility defect in these patients, which appear to be related to the thalamus and the limbic system [3,5].

Finally, we conclude that UCTS might be presently one of the best imaging techniques available for the medical diagnosis of EHS and/or MCS patients.

## Figures and Tables

**Figure 1 diagnostics-10-00427-f001:**
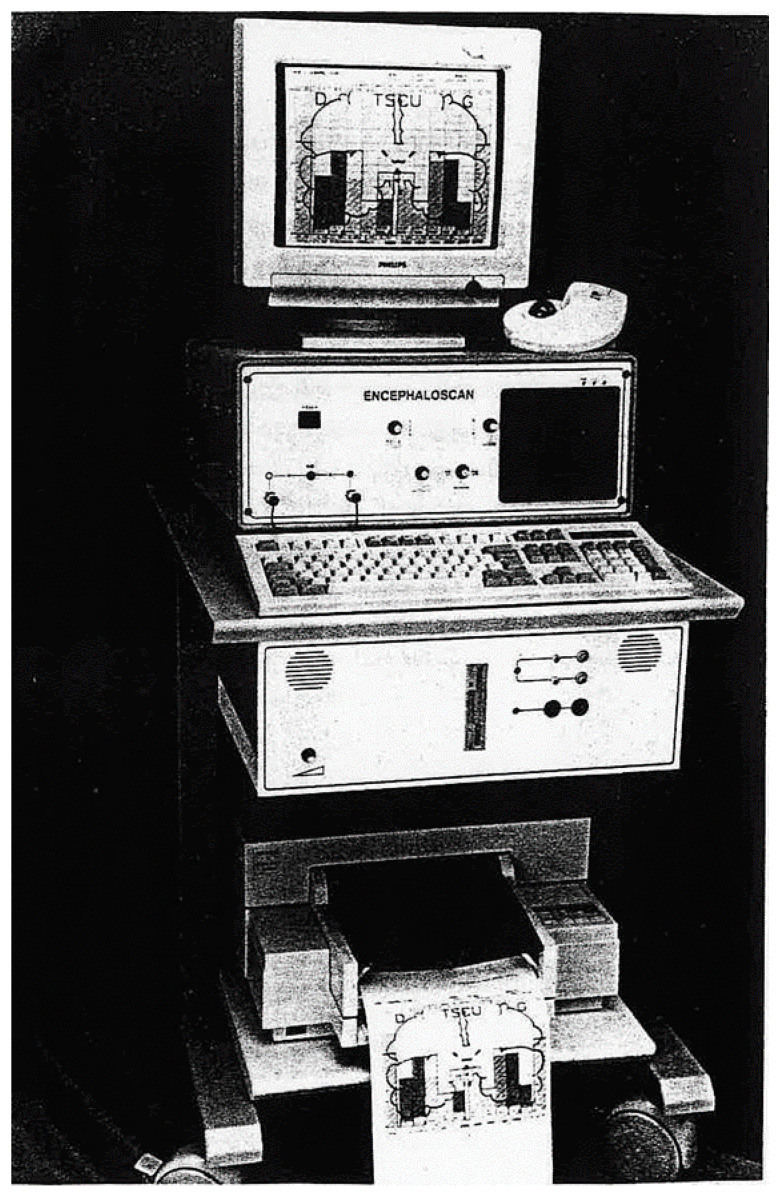
Echotomosphygmograph prototype still in use in 2018 in Paris, on which clinical research can be performed.

**Figure 2 diagnostics-10-00427-f002:**
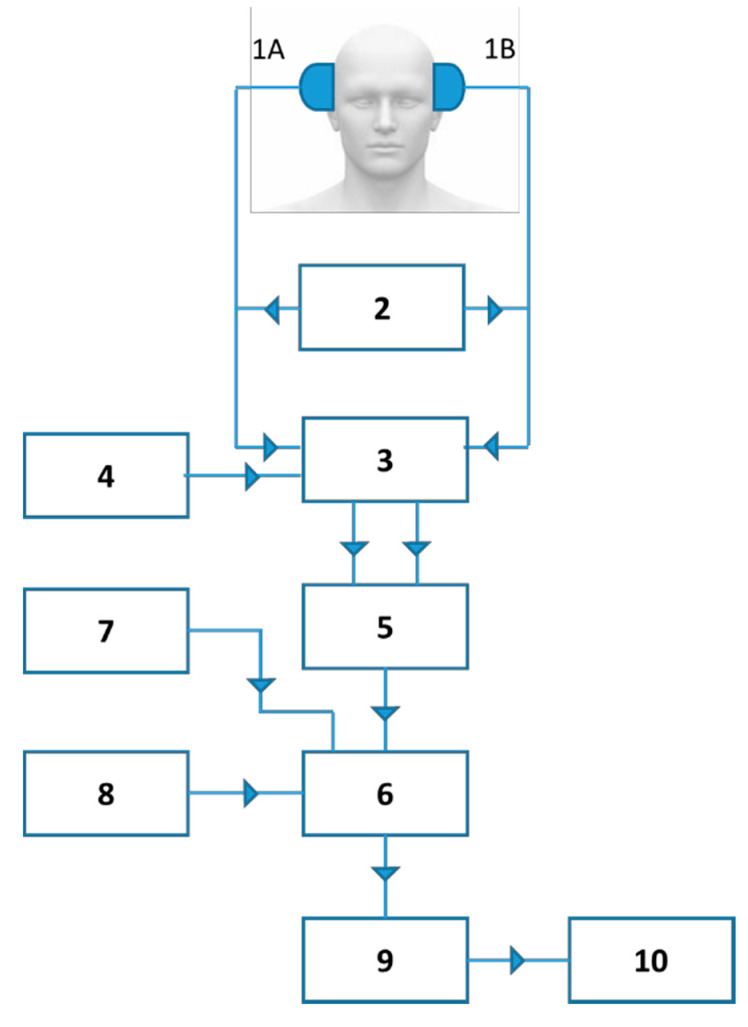
Technological arrangement concerning the equipment used (according to reference 27): 1A, 1B, 2-MHz emitting–receiving probes; 2, voltage transmitter; 3, Amplifier; 4, Time gain compensator (TGC); 5, Binary conversion system (frequency of 18.8 MHz); 6, Ultrafast computing unit; 7, Delay device; 8, Digital rejection filtering system; 9, Digital computer; 10, video screen (Oscilloscope).

**Figure 3 diagnostics-10-00427-f003:**
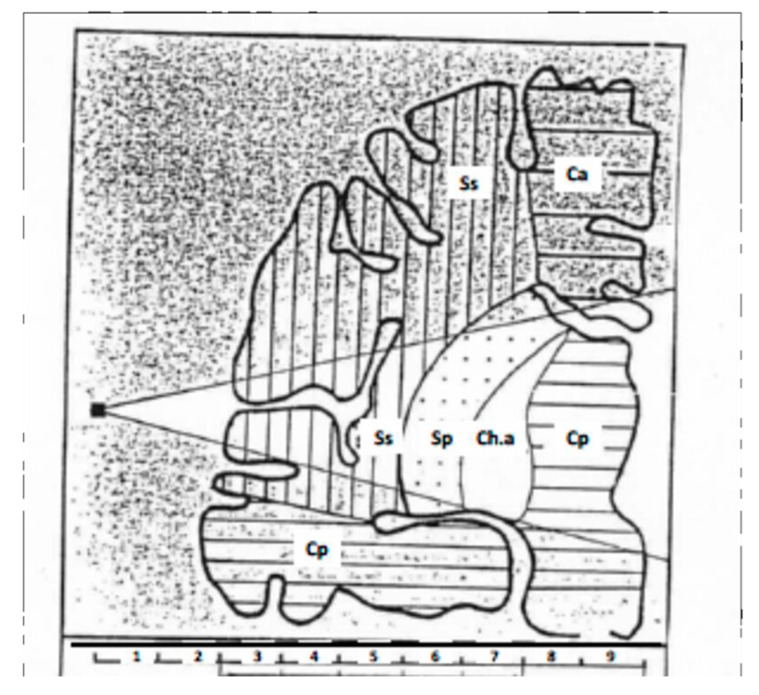
Hypothesis regarding the different neuro-vascular temporal territories included, from surface to depth, in the ultrasound field [26]. Ca, anterior cerebral artery territory; Cp, posterior cerebral artery territory; Ch.a, anterior choroidal artery territory; Sp, deep Sylvian territory; Ss, Superficial Sylvian territory.

**Figure 4 diagnostics-10-00427-f004:**
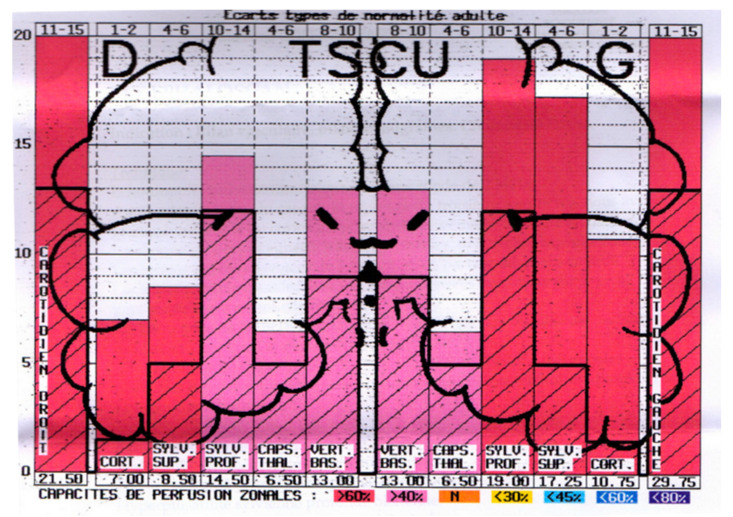
Example of pictogram showing the normal mean pulsometric index (MPI) values in the different regions individualized by UCTS in both temporal lobes (dark line) from the surface (cortex) to the median line of the brain wall of the third ventricle and real values of the pulsometric index (PI) for each region recorded in this particular example [3]. Note that here PI are over normal MPI in all regions.

**Figure 5 diagnostics-10-00427-f005:**
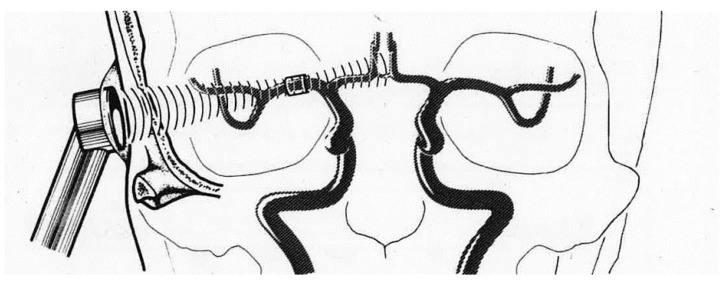
Measurement of blood flow velocity in the middle cerebral artery by transcranial Doppler [29].

**Figure 6 diagnostics-10-00427-f006:**
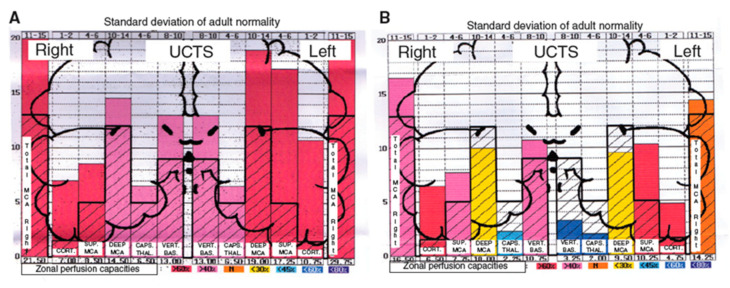
A case and a control using UCTS to explore the global centimetric ultrasound pulsatility in the two temporal lobes of: a normal subject (**A**); and an EHS self-reporting patient (**B**) [3]. Measurements are expressed in Pulsometric Index (PI). Note that in Figure 6A,B mean values of PI in each explored area recorded are from the cortex to the internal part of each temporal lobe; thus, on the left part of the two diagrams Figure 6A,B for the right lobe, mean PI values are expressed from the left to the right, while, on the right part of these diagrams for the left lobe, mean PI values are expressed from the right to the left. Note that in Figure 6A (normal subject) all values are over the normal mean values, whereas in Figure 6B (EHS-self reporting patients) values in the so-called capsulothalamic areas (the fifth and second columns for the right and left temporal lobe, respectively) are under the normal mean values.

**Table 1 diagnostics-10-00427-t001:** The different individualized regions in each of the temporal lobes from surface to depth and their clustering according to their correspondence with the different vascular and neurological structures they are associated with.

Centimeters 3 and 4	Cortico-subcortical region
Centimeters 3–5	Superficial Sylvian region
Centimeters 5–7	Deep Sylvian region
Centimeter 7	Intermediate region (capsulo-thalamic)
Centimeters 3–7	Carotid region

**Table 2 diagnostics-10-00427-t002:** Estimates normal values of mean pulsometric indexes (MPI) in the different region considered [26].

Pulsometric Index	Very High	High	Normal	Slightly Low	Low	Very Low	Extremely Low
**Cortico-Subcortical**	4–6	2–4	1–2	0.5–1	0.25–0.50	0–0.25	0
**Superficial Sylvian**	8–12	6–8	4–6	3–4	2–3	0.50–2	0–0.50
**Deep Sylvian**	18–22	14–18	10–14	6–10	4–6	2–4	0–2
**Global Carotid**	20–25	15–20	11–15	8–11	5–8	3–5	0–3
**Intermediate**	7.5–8.5	6–7.5	4–6	3–4	2–3	1–2	0
**Vertebro-Basilar**	13–16	10–13	8–10	6–8	4–6	2–4	0–2

**Table 3 diagnostics-10-00427-t003:** Mean pulsometric index measurement (±SD) obtained by using UCTS in the different tissue areas investigated in temporal lobes of EHS-, MCS-, and EHS/MCS-bearing patients relative to normal concomitant controls (unpublished data).

Temporal Lobe	Tissue Areas Analyzed	Apparently Healthy Subjects ± SD *n* = 84	EHS Patients ± SD *n* = 65	*p* **	EHS-MCS Patients ± SD *n* = 29	*p* ***	MCS Patients ± SD *n* = 6	*p* ****
**Right**	carotidian	20.39 ± 4.33	12.74 ± 4.25	<0.00001	12.38 ± 3.58	<0.00001	12.75 ± 2.12	0.0004
	cortical-subcortical	6.02 ± 2.90	5.26 ± 3.01	**0.19**	4.91 ± 3.04	**0.12**	4.25 ± 1.68	**0.19**
	superficial MCA	10.38 ± 3.99	7.79 ± 3.55	0.00055	7.97 ± 3.89	0.013	8.55 ± 1.97	**0.32**
	deep MCA	14.40 ± 2.44	7.47 ± 2.78	<0.00001	7.82 ± 2.34	<0.00001	8.50 ± 1.57	<0.00001
	capsulo-thalamic	5.81 ± 0.96	2.14 ± 1.56	<0.00001	2.22±1.57	<0.00001	3.00 ± 0.59	<0.00001
	vertebro-basilar	11.04 ± 1.65	8.02 ± 2.12	<0.00001	8.45 ± 1.61	<0.00001	7.55 ± 0.94	<0.00001
**Left**	vertebro-basilar	10.95 ± 1.81	5.22 ± 2.71	<0.00001	5.55 ± 3.04	<0.00001	5.40 ± 3.34	<0.00001
	capsulo-thalamic	5.32 ± 1.38	3.24 ± 2.19	<0.00001	3.19 ± 2.28	<0.00001	3.00 ± 2.21	0.0016
	deep MCA	13.89 ± 2.54	9.77 ± 3.36	<0.00001	9.33 ± 3.19	<0.00001	6.95 ± 3.92	<0.00001
	superficial MCA	11.29 ± 4.25	8.63 ± 3.51	0.0005	8.81 ± 3.61	0.012	7.15 ± 2.99	0.039
	cortical-subcortical	6.88 ± 3.50	5.06 ± 2.93	0.004	5.34 ± 3.25	**0.063**	4.95 ± 3.00	**0.24**
	carotidian	20.58 ± 5.09	14.85 ± 4.55	<0.00001	17.27 ± 13.58	**0.15**	11.90 ± 2.95	0.0005

EHS, electrohypersensitivity; MCS, multiple chemical sensitivity EHS; SD, standard deviation. ** Significance levels (*p* values) obtained by using the two tailed Student t-test for comparison between the EHS-bearing patients and the apparently healthy subjects used as concomitant controls. *** Significance levels (*p* values) obtained by using the two tailed Student t-test for comparison between the EHS-MCS-bearing patients and apparently healthy subjects used as concomitant controls. **** Significance levels (*p* values) obtained by using the two tailed Student t-test for comparison between the MCS-bearing patients and apparently healthy subjects used as concomitant controls. Bold values indicates no-statistical significance.

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
