# Peer review of "Technical Assessment of Ultrasonic Cerebral Tomosphygmography and New Scientific Evaluation of Its Clinical Interest for the Diagnosis of Electrohypersensitivity and Multiple Chemical Sensitivity"

_diagnostics, 2020, doi:10.3390/diagnostics10060427_

Round 1

Reviewer 1 Report

This is an extracranial measurement, not an intracranial, so there is clearly an assumption that the temporal lobes are located beneath the sensors.

Fig. 4 and 6 shows an illustration of a brain, not showing the exact anatomy of an individual, i.e., no anatomical information is acquired through this modality.

Reviewer 2 Report

Revision of the paper is sound, the aim of the study presented is now clear. I'm still standing on my point of view, that ultrasound trajectory is of utmost importance, because the anatomical variations (so ultrasound speed and pulsations) might differ not only in different patients, but in right and left side of the brain of the same patient. But explanation of author is convincing, making my criticism a scientific discussion. From my point of view, no further corrections of the text are necessary.

This manuscript is a resubmission of an earlier submission. The following is a list of the peer review reports and author responses from that submission.

Round 1

Reviewer 1 Report

This paper overviews the imaging using ultrasonic cerebral tomosphygmography (UCTS), which is a historical device for investigating the brain.  The author introduces the device, measurement system and studies of using this technique.

This is an interesting review for learning an aspect of the neuroimaging history.  However, the current neuroimaging is based on the fusion of both anatomical and functional information obtained by multimodal approach, whereas UCTS uses simplified 2-D illustration for anatomy and only measures the pulsatility in the temporal area of the head (not brain).

Reviewer 2 Report

Ultrasonic Cerebral Tomoshpygmography is not a new method, and is described in the literature.

No new information is provided - the priciples of method are known, no additional anatomical information is presented, conclusions are drown from the theoretic anatomical knowledge and is not compartable with ourdays individualized capabilities of neuroradiology.

The distiction of transcranial Doppler and UCTS is emphasized few times in the paper, however the comparison is presented in Part 6?

Consider changing the aims and the structure of the study.

Reviewer 3 Report

This review by F. Greco talks about Ultrasonic encephaloscan as a tool for the diagnosis of electrohypersensitivity and various chemical chemical sensitivities. Of late, few scientists are proposing the revival of this obsolete technique. One of the papers that was missing from the citation is by Irigaray, et al., J Clin Diagn Res 2018, 6:1. This is important as it iss one of the most recent article.

My first biggest criticism about this review that it is lacking citation for the figures. Of course, they are borrowed and hence, one cannot write reviews and publish figures without citing it properly. Second, it does not say anything new than what is out there.  I would really appreciate if you could provide the mechanism behind it i.e. the physics and the engineering. That is lacking in most reviews as most people don't understand how it works. That kind of review would help revive the obsolete technique as it cannot make a comeback from the dumpster. It has to be adapted and reformed using modern engineering and science and then it can make a comeback.

Overall, I did not not find anything new and there are articles which are really similar. Please add something to this review that would champion the cause of reviving an obsolete technique.